# Family caregivers' perspectives on the acceptability of four interventions proposed for rural transitional care: A multi-method study

**Mary T. Fox**[1,2]*, **Jeffrey I. Butler**[1,2], **Souraya Sidani**[3], **Manal M. Alzghoul**[4¤], **Mark Skinner**[5,6], **Travis Amell**[1,7], **Mary Ferguson-Paré**[1]

1 School of Nursing, York University, Toronto, Ontario, Canada, 2 York University Centre for Aging Research and Education, Toronto, Ontario, Canada, 3 Faculty of Community Services, Ryerson University, Toronto, Ontario, Canada, 4 School of Nursing, Lakehead University, Thunder Bay, Ontario, Canada, 5 Trent School of the Environment, Peterborough, Ontario, Canada, 6 Trent Centre for Aging & Society, Peterborough, Ontario, Canada, 7 Hospice of Waterloo Region, Waterloo, Ontario, Canada

¤ Current address: Department of Nursing, Brock University, St. Catharines, Ontario, Canada
* maryfox@yorku.ca

## Abstract

### Background

There is a critical need for hospital-to-home transitional care interventions to prepare family caregivers for patients' post-discharge care in rural communities. Four evidence-based interventions (named discharge planning, treatments, warning signs, and physical activity) have the potential to meet this need but family caregivers' perspectives on the acceptability of the interventions have not been examined. This gap is significant because unacceptable interventions are unlikely to be used or used as designed, thereby undermining outcome achievement. Accordingly, this study examined the perceived acceptability of the four interventions to rural family caregivers.

### Materials and methods

A multi-method descriptive design was used. The quantitative method entailed the administration of an established scale to assess the interventions' perceived acceptability to family caregivers. The qualitative method involved semi-structured interviews to explore family caregivers' perceived acceptability of the interventions in greater depth, including acceptable and unacceptable aspects, in the context of their own transitional care experience. Participants were the family caregivers of a relative who had been discharged home in a rural community from an acute care hospital in Ontario, Canada.

### Results

The purposive sample included 16 participants who were mostly middle-aged women (n = 14; 87.5%) caring for a parent (n = 9; 56.3%) at high risk for hospital readmission. The mean scores on the acceptability measure were 3 or higher for all interventions, indicating that, on

**Data Availability Statement:** According to the guidelines of our hospital site's Research Ethics Board, we are not permitted to post the study data for public consumption. Given that our sample was

recruited from small towns and participants narratives may be recognizable to others even in de-identified data, we are also unable to share the data. However, researchers interested in using our data may submit a request to the hospital REB to request use of the data. Contact info: Research Ethics Board Health Sciences North Research Institute 56 Walford Road Sudbury, ON, P3E 2H2 Phone: 705-523-7300 Email: reb@hsnsudbury.ca

**Funding:** This study was supported by the Ontario Ministry of Health & Long-Term Care (https://www.health.gov.on.ca/en/), Health System Research Fund, grant #484 (awarded to MTF). The funders had no role in study design, data collection and analysis, decision to publish, or preparation of the manuscript.

**Competing interests:** The authors have declared that no competing interests exist.

average, the four interventions were perceived as acceptable. In terms of acceptable aspects, four themes were identified: the interventions: 1) involve family caregivers and pro-actively prepare them for discharge, 2) provide clear, written, and detailed guidance, 3) place the onus on healthcare providers to initiate communication, and 4) ensure post-discharge follow-up. In terms of unacceptable aspects, one theme was identified: the physical activity intervention would be challenging to implement.

## Discussion

The findings support implementing the four interventions in practice throughout the hospital-to-home transition. Healthcare providers should assess family caregivers' comfort in partici-pating in the physical activity intervention and tailor their role accordingly.

## Introduction

The transition from hospital-to-home is a high-risk event in a patient's recovery. Approxi-mately 28% of North American patients experience complications [1] and between 9.4% and 11.6% are readmitted to hospital within 30 days of discharge [2, 3]. Patients in rural communi-ties in Canada are particularly prone to these problems due to their older age, high prevalence of chronic conditions, lower socio-economic status, and inadequate healthcare resources, leav-ing many highly reliant on their families for post-discharge care [4]. Consequently, there is a critical need for hospital-to-home transitional care interventions to adequately prepare family caregivers (FCs), defined as the primary caregiver of a patient, for post-discharge care in rural communities.

Four evidence-based interventions (named discharge planning, treatments, warning signs, and physical activity) have the potential to prepare FCs for post-discharge care. However, the interventions were largely designed and tested in urban settings and may not address the needs of rural FCs. There is limited understanding of rural FCs' perspectives on their accept-ability i.e., the desirability of an intervention in addressing a health problem or need [5]. This is a significant gap because interventions that are not acceptable to FCs are unlikely to be used or used as designed, thereby undermining outcome achievement [5, 6] and exacerbating the challenges of rural TC [4].

To fill this knowledge gap, this study examined the perceived acceptability of the four evi-dence-based interventions to rural FCs. Engaging FCs in assessing the interventions' accept-ability prior to their implementation may help identify areas requiring modifications, thus ensuring that the interventions are responsive to FCs' needs and relevant to their context [7]. Furthermore, providing acceptable interventions reduces wasted efforts and minimizes costs [8], which is particularly important in rural communities given their limited resources [9].

### Evidence-based interventions proposed for rural transitional care

The four evidence-based interventions (hereafter referred to as "interventions") proposed for rural TC were designed to address the post-discharge care management needs of adult patients and their FCs [10]. Evidence of the interventions' efficacy in promoting health and wellbeing, preventing complications, and reducing hospital readmissions was synthesized from previous experimental and quasi-experimental studies and literature reviews [11–17]. The interventions are delivered in person with follow-up via telephone or home visit sessions. The interventions

are offered during hospitalization and, apart from discharge planning, continue over the 30-day period following discharge, as needed. All the interventions incorporate verbal educational strategies to promote learning, including the teach-back method as well as written materials in lay language (S1 File).

**Discharge planning.** Discharge planning sets the stage for the other interventions. Its main goal is to prepare patients and FCs to manage patient care at home. Its key components include: (1) assessment of patient and FC needs related to care management within the immediate post-discharge recovery period; (2) involvement of patients and FCs in setting goals and in planning patient care; (3) provision of education on post-discharge care management and opportunities to practice relevant care-related skills; and (4) provision of support to address barriers to care management.

**Treatments.** The treatments intervention aims to enhance patients' and FCs' knowledge, confidence, and ability to apply pharmacological (e.g., medications) and non-pharmacological (e.g., wound care) treatments correctly. It involves the: (1) identification and resolution of any discrepancy in the prescribed treatments; (2) assessment of patient and FC learning needs related to understanding the prescribed treatments (what they are, why they are needed) and their correct enactment; (3) involvement of patients and FCs in identifying their learning needs and strategies to meet those needs; and (4) provision of opportunities to practice, how to carry out the selected strategies.

**Warning signs.** The goal of the warning signs intervention is to inform patients and FCs of signs and symptoms indicative of worsening health conditions. It consists of the: (1) assessment of patients' and FCs' learning needs about warning signs that are relevant to the patients' specific health condition; (2) provision of information about the warning signs (what they are, how to recognize and monitor them) and about strategies to address the warning signs including when to contact healthcare providers and go to the emergency department.

**Physical activity.** The physical activity intervention aims to promote patient engagement in safe physical activity, prevent functional decline, and promote return to usual daily activities. Its main components are the: (1) assessment of patients' mobility and need for assistive devices; (2) involvement of patients and FCs in establishing physical activity goals and monitoring related progress, identification of physical activity barriers and strategies to address them; (3) provision of education on rest, safe mobility, and physical activities to perform (based on the patient's condition), how, where, when, and how often.

## Materials and methods

### Research questions

1. What are the perspectives of FCs on the acceptability of the four interventions in meeting their needs?

2. Which aspects of the interventions are acceptable/unacceptable and why?

### Study design

This study employed a multi-method descriptive design. The quantitative method entailed the administration of an established scale to assess the interventions' perceived acceptability to FCs [18]. The qualitative method involved semi-structured interviews to explore FCs' perceived acceptability of the interventions in greater depth in the context of their own TC experience.

**Sample and setting.** Our purposeful sample comprised English-speaking and reading adults aged 18+, who were the unpaid, primary caregivers of rural-dwelling adults admitted to hospital for a surgical procedure or medical illness, at risk for hospital readmission, and discharged home. Participants were recruited through hospital referrals and flyers posted in healthcare institutions in rural Southwestern and Northeastern Ontario. We recruited 16 FCs, which was sufficient to achieve informational saturation as determined independently by two team members (MTF and JIB).

## Data collection

Data collection was conducted via telephone using quantitative measures and a qualitative semi-structured interview guide (S2 File). Quantitative measures included FC eligibility screening measures, socio-demographic measures, and measures of intervention acceptability, described below. Prior to the interviews, FCs were mailed a package containing the measures and a lay description of each intervention, which FCs were invited to follow along with during data collection.

The interviews were conducted one-on-one by a doctoral trained Research Associate (RA) with over fifteen years of experience conducting qualitative interviews. The RA (JIB), who self-identifies as a white, cis-gendered man, introduced himself to participants as a sociologist who wanted to understand their hospital and post-discharge experience. As a highly educated researcher, the RA was conscious of his socio-economic privilege relative to participants, and continuously reflected on how this may shape his interactions with them.

In the interview, FCs were prompted to describe their TC experience to identify their needs (e.g., "Tell me about the parts of your relative's care that you feel you were unprepared for and found hard to manage at home?") as a segway to determine the relevance of the interventions in meeting their needs in managing their relative's post-discharge care. The RA then described each intervention and administered the measure assessing each intervention's acceptability. We randomized presentation of the measures and their descriptions to control for possible order effects. After FCs responded to each intervention acceptability measure, they were engaged in a discussion about the potential of the interventions to meet their needs and aspects of the interventions that were acceptable or unacceptable (e.g., "What is it about this intervention that you do not find acceptable? You indicated that you would be unlikely to use or participate in this intervention. Can you tell me why?"). Interviews lasted approximately one hour and were audio-recorded and transcribed. Field notes were taken following the interviews. Data collection was completed within 30 days after FCs' relatives were discharged from hospital. To reduce burden, transcripts and findings were not returned to FCs for verification.

**Research ethics.** Ethics approval was obtained from the Office of Research Ethics at York University, Certificate#: e2018-014, and from the Research Ethics Office at Health Sciences North Research Institute, Project# 18–053. Written informed consent was obtained from all participants. All methods were carried out in accordance with the relevant guidelines and regulations of the World Health Organization's Declaration of Helsinki.

**Screening and socio-demographic measures.** The LACE index was used to determine if participants were caring for a relative who was at risk for hospital readmission. The index utilized four variables to predict risk for hospital readmission during the 30-day post-discharge period: length of hospital stay ("L"), acuity of hospital admission ("A"), comorbidities ("C") and ED visits in the 6 months before admission ("E") [19]. LACE data were obtained by report from FCs recruited through study flyers and by extraction from the medical record for FCs recruited at hospitals. LACE indices between 5 to 9 indicate moderate risk of hospital

readmission while indices > 9 indicate high risk [19]. Psychometric properties of the LACE are reported elsewhere [20].

Eligible participants were the relative, friend, or significant other and self-identified as the primary caregiver of a patient discharged home to a rural community. To ascertain if participants were caring for a relative in a rural community, we employed the Rurality Index of Ontario–a census-based metric, derived from population size and travel time to the closest healthcare facility [21]. Indices were acquired by entering the patient's postal code into an online calculator; indices > 40 indicate that the FC is providing care to a relative in a rural community and greater indices indicate greater rurality [22]. Other eligibility criteria (e.g., age) were determined with self-report measures. FCs' socio-demographic profile (e.g., level of education) was measured by standard self-report questions.

**Intervention acceptability measure.**   The Intervention Acceptability Scale is a self-report scale that measures perceptions of intervention acceptability. The scale provides a written description of each intervention followed by a set of items to rate its acceptability in terms of its appropriateness, effectiveness, risks, and ease of use. The scale was designed to be contextualized for different health interventions [18]. For example, some items require the researcher to specify the outcome expected of the intervention. To illustrate, item 3 for the scale for the discharge planning intervention asked: How effective do you think this intervention would be in *preparing you to manage your relative's care and recovery at home after a hospital stay*? A five-point scale ranging from *not at all* (0) to *very much* (4) was used in the rating. The total score was derived by taking the mean of the items' scores to reflect participants' perceived overall acceptability of the interventions. A five-point scale ranging from *not at all* (0) to *very much* (4) was used in the rating. The total score was derived by taking the mean of the items' scores to reflect participants' perceived overall acceptability of the interventions. The scale has demonstrated internal consistency reliability (alpha > .80) [23–26] as well as content, discriminant [25, 26] and factorial validity (factor loadings > .30) [25] in prior research on different health interventions and populations. In the larger study, the scale had Cronbach alphas ranging from 0.85 to 0.89 across the four interventions [27].

## Data analysis

Descriptive statistics, in accordance with each variable's level of measurement, were used to describe FCs' average standing and variation on all measures using SPSS. Interventions that were acceptable had a mean rating > 2 (i.e., response scale midpoint) and FCs' narratives indicated that the interventions have the potential to meet their needs [28]. Interview transcripts were coded and analyzed independently by two researchers (MTF and JIB) using conventional qualitative content analysis with NVivo (Version 11) to manage the data. Analysis focused on identifying aspects of the interventions FCs perceived as acceptable or unacceptable and meeting or not meeting their needs, and the reasons why [29]. This process involved inductively developing preliminary codes, iteratively synthesizing them into hierarchical categories, which were then developed into themes [30]. Each code, category, and theme were defined, and their interconnections documented. During the analysis, any areas of disagreement of the coding authors were debated until intersubjective consensus was achieved. MTF brought her viewpoint as a nurse and JIB as a sociologist to the discussion. Notes were maintained of all decisions made and their rationale. Strategies for trustworthiness were employed [31]. For example, dependability was actualized by having all team members review the findings and then discussing their cogency and how their presentation could be further refined or enhanced.

## Results

### Sample

Fifty-six FCs expressed interest in the study but 6 declined after having heard what participation would entail and another 22 did not meet the eligibility criteria. The remaining 28 FCs consented but two became ineligible because their relative's status changed and 10 withdrew (3 explicitly due to competing demands and 7 implicitly when they failed to respond to interview scheduling requests). The final sample included 16 FCs.

**Family caregiver demographic characteristics.** All FCs identified as white and as a family member (e.g., daughter) of their relative and most identified as women (n = 14; 87.5%). Their median age was 49 years (range 26–67); most were common law or married (n = 11; 68.8%) and employed outside of the home full-time (n = 10; 62.5%). Community college diploma, trades certificate or apprenticeship was their highest level of education (n = 10; 62.5%). The majority were caring for a parent (n = 9; 56.3%) and lived with the relative they were caring for (n = 10; 62.5%). The average FC was providing care to a relative with a median age of 69 years (range 20–87), living in a community with a median rurality index of 50 (range 41–89), admitted to hospital for a medical illness (n = 12; 63%) and had a high risk of hospital readmission, designated by a mean LACE index of 10.9 ($\pm$ 2.7).

### Family caregiver perspectives on intervention acceptability

**Quantitative findings of intervention acceptability.** As reported in Table 1, the mean total scores on the Treatment Acceptability Scale were 3 or higher, which is above the mid-range of the scale (i.e., 2), for all interventions, indicating that, on average, FCs perceived all four interventions as acceptable. The mean scores for each subscale were also above 3, indicating that, on average, FCs perceived the interventions as appropriate, effective, and convenient with minimal risk.

**Qualitative findings of intervention acceptability.** In general, FCs perceived the four interventions as acceptable and having the potential to meet their needs. FCs were highly laudatory of the interventions and made positive comments such as "this whole intervention system sounds ideal", "it would be a great program", "it would be very valuable, very, very useful. . .to educate somebody before they go home" and "I would love nothing more than to see a process like this come into practice". FCs underscored the potential value of initiating the interventions in hospital and continuing them after discharge. As one participant noted, "it [would be] an excellent service to have because people aren't staying in the hospital as long. So, families have to take care of their relatives more so than they have in the past".

**Table 1. Descriptive results of participants' acceptability ratings of the interventions.**

|  | Discharge planning | Warning signs | Treatments | Physical activity |
|---|---|---|---|---|
|  | M (SD) | M (SD) | M (SD) | M (SD) |
| **Scale** (range of values) |  |  |  |  |
| Overall Acceptability (0–4) | 3.44 (0.48) | 3.75 (0.34) | 3.58 (0.50) | 3.46 (0.62) |
| **Subscale** (range of values) |  |  |  |  |
| Appropriateness (0–4) | 3.44 (0.85) | 3.75 (0.45) | 3.59 (0.82) | 3.56 (0.89) |
| Effectiveness (0–4) | 3.31 (0.77) | 3.81 (0.31) | 3.59 (0.49) | 3.63 (0.50) |
| Ease of Use (0–4) | 3.47 (0.50) | 3.66 (0.54) | 3.56 (0.48) | 3.06 (1.14) |
| Risks (0–4) | 3.69 (0.73) | 3.81 (0.75) | 3.56 (0.89) | 3.56 (1.09) |

*Note*. Repeated measures analysis of variance identified no statistically significant differences on scale and subscale acceptability scores amongst the four interventions.

*Acceptable aspects of the interventions*. FCs' narratives focused on four key aspects of the interventions' acceptability: the interventions 1) involve FCs and proactively prepare them for discharge, 2) provide clear, written, and detailed guidance, 3) place the onus on healthcare providers to initiate communication, and 4) ensure post-discharge follow-up.

## 1. The interventions involve FCs and proactively prepare them for discharge

FCs valued the "proactive" nature of the interventions. They saw major benefits in healthcare providers beginning to assess patient and FC needs early on and then preemptively planning and working with them to ensure that they were prepared for a successful transition home. As FC5 expressed:

> just the fact that within 24 hours they [healthcare providers] start to come up with a plan for discharge, to be ahead of the game and get things in order. . .gives you, the family member, peace of mind to know that those things are getting put in place.

This contrasted sharply with FCs' own TC experiences of having received preparation for managing post-discharge care "on the way out the door", if at all, which left them in need of more information about their relative's care.

In comparing her own experience to the interventions, FC14 explained that she had not been involved in discharge preparation "so I didn't get all that great information that I could have used." Other FCs reflected on how they were expected to carry out discharge plans that they had not been engaged in developing. For example, FC1 recalled how she was not involved in the preparatory work needed to send her 82-year-old mother home with oxygen.

> Once she [FC1's mother] was released, then we had to go to her home. They [healthcare providers] gave us a portable tank for her to get home and then we had to wait for a lady to come and set up the unit. . . .I was working from 8:30 [a.m.] and now I'm dealing, right after work, to get all this information about this oxygen machine and her portable stuff. . .It would have been helpful for me to get prepared in some ways. Like, I didn't know how big this unit was. It takes a lot of room. If I could have went home and prepared that prior to her getting released and understanding what it all means.

## 2. The interventions provide clear, written, and detailed guidance

FCs valued the step-by-step guidance the interventions provide. FC15 noted that "the biggest positive I see here is clear instructions. . .everything is written out with clear instructions versus being told [as he reflected on his own TC experience of having received vague instructions] 'by the way, look for a fever', 'come back if it gets worse'." Other FCs contrasted this clarity with the imprecise directives, such as 'walk but take it easy", they had received and criticized as too unclear and lacking in specificity to be of any real value to them. FC8 explained that written, detailed guidance would have helped her because she was not always "100% certain if I'm doing what I'm supposed to be doing".

FCs expressed that such clear guidance would have provided them the opportunity to learn about their relative's care and how to best support it. When comparing the interventions to her own TC experience, FC14 noted that:

> being educated about what to expect and what to do is really important when taking somebody home [from hospital]. If somebody [healthcare providers] would have given me a

clearer picture of what to expect, I think I would know better instead of making my own assumptions.

This benefit applied to the interventions broadly but was also highlighted specifically for the warning signs intervention. FC11 acknowledged that "I would have loved to have known a little more what some of the things that I should be looking for were, to indicate where he's [FC11's husband] getting worse."

### 3. The interventions place the onus on healthcare providers to initiate communication

FCs valued that the interventions place the onus for initiating communication on the health-care team. FCs liked that healthcare providers would actively seek FCs out to educate them about the specifics that they needed to know to provide post-discharge care. Comparing the interventions to their own TC experience, FCs noted that healthcare providers typically did not initiate conversations with them about post-discharge care. FCs conceded that although they tried to ask questions and solicit details about their relative's post-discharge care, they were unaware of what they needed to know and ask about explicitly, and thus did not inquire about or seek clarification about issues they only realized were important once home. FC1 recounted that she felt as though the impetus for initiating conversations about her mother's post-discharge care fell squarely on her. She lamented that "the nurses only answer what you ask them" and there was "no communication [in hospital] unless I ask questions". FC10, was similarly frustrated that, despite spending long hours at the hospital during her relative's stay, she found it very difficult to communicate with healthcare providers.

### 4. The interventions ensure post-discharge follow-up

FCs, like FC5, highlighted that the interventions' inclusion of nurse follow-up within 48 hours of discharge and over the next few weeks was a "very big plus" because it provided much needed reassurance. FCs expressed that the first few weeks after discharge were particularly stressful, during which they felt "uncertain" and "nervous" about their relative's post-discharge care and mismanaging it, potentially resulting in complications. As FC5 explained while contemplating how the interventions would have altered her own experience, the follow-up described in the interventions would have:

taken the worry off from me, because then at least someone was looking and keeping an eye on things, and [I] wasn't trusting my eyes to pick something up that I might not recognize.

FCs viewed follow-up as having the potential to provide the assurance that they were delivering care correctly and identify and address their knowledge needs. As FC4 articulated, "having them [a nurse] follow up just gives you that bit of confidence that there is somebody out there, that you're not in there alone." FC4 elaborated that having a nurse provide follow-up would have been helpful in:

guiding you through, or if they [FCs] have a question they [nurses] can answer it. You don't have to sit here and wonder if I'm doing the right thing or not.. . .just having that reassurance, having somebody that you can talk to about it. . .just making me feel more sure about what *I'm doing*, and more sure about what *I should be doing* . . .Because you're not sitting here trying to figure out how to do it.

This contrasted sharply with the experience of most FCs, who felt that they had to figure out care on their own.

*Unacceptable aspects of the interventions.* Because the interventions were overwhelmingly perceived positively, participant comments about unacceptable intervention aspects were scant. Some FCs identified problematic areas specific to the physical activity intervention.

## 1. The physical activity intervention would be challenging to implement

Some FCs explained that it would be difficult for them to use the physical activity intervention. They foregrounded numerous practical considerations in this vein. For instance, FCs asserted that this intervention "would be a little bit time consuming and maybe a little harder to follow through on" (e.g., tracking number of steps taken), that helping a larger relative mobilize may not possible, or that they could not provide the intervention due to work or family responsibilities or because they lived too far away to regularly oversee their relative's physical activity. Other FCs were concerned about inadvertently overstraining their relative or exacerbating their pain.

Additionally, some FCs remarked that they anticipated difficulty getting their relative to adhere to the physical activity intervention and that, because the intervention requires FCs to monitor and reinforce their relative's physical activity, it may lead to conflicts. FC4 maintained that "it's kind of like a child with their parent, you know? Just the whole resistance part." She suggested that her husband's hospitalization led to him feeling disempowered, and consequently asserting himself by refusing to walk. FC4 perceived her husband's resistance as "a control thing, because he doesn't have a lot of control over some things in his life, but he can control whether or not I help him, control him. If he'll think I'm trying to control him he doesn't want that, he wants his own control, autonomy." As a result, these FCs expressed that they did not want to take on the role of promoting their relative's physical activity. Notably, all the FCs who raised this concern were women caring for a male relative who was either their spouse or father, suggesting that gender may be an important factor shaping these dynamics.

## Discussion

This study responds to recent calls for greater scholarly attention to TC in rural communities [32]. Most prior literature on TC is characterized by a distinct urban centrism, which undermines the applicability of TC interventions to rural contexts and their potential health benefits. Our study thus explicitly addresses the urban normativity that underlies current TC interventions by illuminating the perspectives of rural FCs. We identified that FCs in rural Ontario perceived the four interventions as acceptable overall, as well as effective, appropriate, and convenient with minimal risk. Accordingly, these four interventions constitute sound options for rural transitional care. Healthcare providers may use the interventions with FCs in rural communities who, based on the findings, are likely to be amenable to employing them.

The qualitative findings corroborated and enriched the quantitative results. FCs saw major benefits in hospital healthcare providers involving them by preemptively planning and working with them to ensure a successful transition home. Such involvement was perceived as giving FCs the peace of mind that they would be prepared for their relative's discharge. Previous research has underscored the multiple benefits of TC interventions involving FCs. For example, involving FCs in reviewing discharge instructions in hospital has been associated with FCs feeling more engaged in post-discharge care [33] and patients having a better understanding of, and greater adherence to, discharge instructions [34]. However, a systematic review highlighted the importance of engaging FCs throughout the transition, rather than just at hospital discharge, because doing so is associated with better TC outcomes [35]. Beyond the

benefits that involving FCs have for patients and for FCs themselves, there is also evidence that involving FCs in hospital discharge planning and decision-making improves system-level outcomes such as reduced hospital readmissions [36, 37], time to readmission, length of stay at hospital readmission, and post-discharge care costs [37]. Accordingly, interventions need to foster FC involvement throughout patient transitions [35, 38].

The interventions FCs assessed in this study differed markedly from their own TC experience, in which they received very little preparation for discharge; this left them feeling unready for post-discharge care and fearful about mismanaging it. This finding is in line with prior research identifying that FCs often feel unprepared for, and overwhelmed by, their relative's post-discharge care [39–41], yet feel responsible for it [40]. Healthcare providers frequently develop discharge plans requiring FC cooperation, but neglect to solicit their input on the plan's feasibility, resulting in heightened risk and stress when FCs have difficulty implementing it [42]. It is thus understandable that FCs want to be involved in discharge planning and sufficiently prepared for discharge [33, 42]. Similarly, in other studies, FCs have underscored that it is crucial for healthcare providers to anticipate and draw out FCs' needs to prepare them to competently provide post-discharge care [42]. Initiating the four interventions in hospital can address this need because it can help FCs develop the skills needed to address an unanticipated health crisis.

There is growing recognition of the importance of concertedly involving FCs in TC. In the United States, 40 states have adopted the Caregiver Advise, Record, Enable (CARE) Act which formally requires healthcare providers to involve FCs in discharge planning and prepare them to provide post-discharge care [43]. There is no similar law in Canada which is concerning given that there has been a move to shorten hospital stays [44]. In this context, the four interventions have the potential to support this policy shift while addressing the need for high quality rural TC that adequately prepares FCs.

FCs were positive about the step-by-step instructions that the interventions entail. They expressed that the clarity of the interventions would have helped them to know what to expect and how to respond to potential signs indicating that their relative's health conditions may be worsening. This concurs with other studies that found FCs benefitted from receiving detailed written instructions to which they could refer after discharge [33]. Such instructions have been found to facilitate FC learning [45], heighten their ability to monitor their relative's recovery [45], increase their adherence to discharge instructions [42, 45], as well as augment their preparedness, competence, and confidence to provide post-discharge care and address potential complications [42]. Other research has emphasized the importance of FCs having clear, written post-discharge care instructions well in advance of discharge [34] and that, similar to the interventions reviewed in this study, written post-discharge care instructions should supplement, not supplant, verbal ones [34, 46].

Our findings identified that FCs appreciated that the interventions place the onus on healthcare providers to initiate communication. That FCs recognized their need for information but did not know what questions to ask concurs with other studies [47–49]. In a systematic review of the literature of how healthcare consumers enact involvement in care, Murray concluded that many healthcare consumers are receptive to receiving information but do not have the capacity to actively seek out the information they need [41] which may be related to health literacy (i.e., the ability to retrieve, process, and use health information) [50]. Many healthcare consumers have low health literacy and studies have found that those with limited health literacy ask healthcare providers significantly fewer questions than their more health literate counterparts [51]. Although we did not specifically measure health literacy, the low level of formal education amongst our sample, which is reflective of rural populations more generally [4], suggests that health literacy may have been in play. The TC literature has underscored

that ensuring healthcare consumers understand what the patient's main health problem is and what they need to do to address it is one of the most important health literacy best practices [52]. In short, the interventions address health literacy because they place the responsibility for initiating conversations about post-discharge care with FCs on healthcare providers.

Our finding that FCs believed receiving home follow-up would provide much-needed reassurance, address their knowledge gaps, and ultimately boost their confidence, is aligned with other evidence emphasizing the importance of providing FCs with reassurance [53]. For example, follow-up is vital after discharge to reassure FCs as it allots the time needed to review instructions that may not have been fully absorbed or to address new issues that may have arisen since discharge [33]. In a qualitative study exploring FCs' perspectives on post-discharge follow-up, FCs who received follow-up reported feeling more secure and prepared in their caregiving, and better able to provide post-discharge care [54]. In a systematic review and meta-analysis, post-discharge follow-up was also found to be effective in reducing hospital readmissions [35]. Follow-up for the FCs of patients at risk for hospital readmission in rural communities is particularly crucial because these communities have minimal formal supports in place and patients in these communities have worse health status and lower primary and specialty health care use than their urban counterparts [55].

The follow-up aspect of the interventions was a departure from the experience of most FCs, who expressed that they had to figure out care on their own. This is significant because other studies have identified follow-up support as FCs' most salient unmet need [56]. The follow-up aspect of the interventions can assuage FCs' fears about complications related to their relative's health conditions; prior research has found that such fears are exacerbated by being far from medical help [57]. These worries are warranted given that 28% of medical patients experience complications in the post-discharge period [1] and the risk of dying of a preventable and treatable condition increases with remoteness [58].

Some FCs qualified that the physical activity intervention stood out as potentially more burdensome than the other interventions and would be difficult to adhere to. Some of FCs' qualms with the physical activity intervention–that it is too time consuming and assumes FC proximity–appear to be novel findings and speak to important dimensions of rural TC, such as distance. Others, such as concerns about patient overexertion and pain exacerbation, managing work and family responsibilities, and sparking interpersonal conflict, have been established in the literature [59–61]. Prior research, for example, has found that FCs were hesitant for their ill hospitalized relative to mobilize because of potential health dangers [62] and FCs experienced competing family and work demands in managing their relative's physical functional needs after hospital discharge [45]. In using the physical activity intervention, healthcare providers should assess FCs' comfort reinforcing patient physical activity and tailor FCs' role accordingly.

In terms of the physical activity intervention having the potential to spark interpersonal conflict, such tension may aggravate pre-existing FC-patient conflictual relationships marked by disagreements surrounding care and decision-making [60]. Role reversal alters power dynamics, especially in situations where the FC was previously in a subordinate position [63]. It is possible that FCs in our study had such conflicts in mind when they expressed hesitancy about overseeing their relative's physical activity. Although further research is needed, that all FCs who raised this concern were women caring for a male relative, suggests that gender may be an important factor shaping these dynamics. Given that our sample was predominantly female, further research is needed to better understand if male FCs share similar perspectives of the physical activity intervention.

In terms of limitations, the study was conducted in one province in Canada and the findings may thus not be generalizable or transferable to all rural communities. The findings may

not be generalizable to racially/ethnically minoritized groups or non-English speakers. Although the Intervention Acceptability Scale did not manifest ceiling effects in prior research [25], it is possible that the scale demonstrated this effect in the current study. Yet, overall, our finding that FCs perceived the four interventions as acceptable supports implementing them in practice or, if there is uncertainty about their effectiveness in rural communities (where prior testing has been limited), moving to future studies evaluating their effectiveness [64]. Future research is needed to examine the acceptability of the interventions to FCs in other rural jurisdictions across Canada and elsewhere, as well as the transferability of FC experiences to different types of rural communities–from the more to less remote.

## Supporting information

**S1 File. Appendix A: Intervention lay summaries.**
(DOCX)

**S2 File. Appendix B: Semi-Structured interview guide for families.**
(DOCX)

## Acknowledgments

The authors are grateful to the family caregivers who shared their perspectives with us.

## Author Contributions

**Conceptualization:** Mary T. Fox, Souraya Sidani.

**Data curation:** Mary T. Fox, Jeffrey I. Butler.

**Formal analysis:** Mary T. Fox, Jeffrey I. Butler.

**Funding acquisition:** Mary T. Fox, Jeffrey I. Butler, Souraya Sidani.

**Investigation:** Mary T. Fox.

**Methodology:** Mary T. Fox, Jeffrey I. Butler, Souraya Sidani.

**Project administration:** Mary T. Fox, Jeffrey I. Butler.

**Resources:** Mary T. Fox.

**Software:** Mary T. Fox, Jeffrey I. Butler.

**Supervision:** Mary T. Fox.

**Validation:** Mary T. Fox.

**Visualization:** Mary T. Fox.

**Writing – original draft:** Mary T. Fox, Jeffrey I. Butler, Souraya Sidani.

**Writing – review & editing:** Mary T. Fox, Jeffrey I. Butler, Souraya Sidani, Manal M. Alzghoul, Mark Skinner, Travis Amell, Mary Ferguson-Paré.

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
