## [Decision Letter · Decision Letter 0]

22 Sep 2022

PONE-D-22-12531Family caregivers’ perspectives on the acceptability of four interventions proposed for rural transitional care: A multi-method studyPLOS ONE

Dear Dr. Fox,

Thank you for submitting your manuscript to PLOS ONE. After careful consideration, we feel that it has merit but does not fully meet PLOS ONE’s publication criteria as it currently stands. Therefore, we invite you to submit a revised version of the manuscript that addresses the points raised during the review process.

Two reviewers have now evaluated your submission. They are both positive about the manuscript but have identified some points that need to be addressed carefully in a revision. Please refer to their comments below for further detail.

We look forward to receiving your revised manuscript.

Kind regards,

Jamie Males

Editorial Office

PLOS ONE

Journal Requirements:

Reviewers' comments:

Reviewer's Responses to Questions

**Comments to the Author**

1. Is the manuscript technically sound, and do the data support the conclusions?

Reviewer #1: Yes

Reviewer #2: Yes

2. Has the statistical analysis been performed appropriately and rigorously? 

Reviewer #1: Yes

Reviewer #2: I Don't Know

3. Have the authors made all data underlying the findings in their manuscript fully available?

Reviewer #1: Yes

Reviewer #2: No

4. Is the manuscript presented in an intelligible fashion and written in standard English?

Reviewer #1: Yes

Reviewer #2: Yes

5. Review Comments to the Author

Reviewer #1: A very well written readable paper on a topic that is very important. Hospitals should not be discharging rural patients without making sure that Family Caregivers are comfortable with their role and the expectations.

Abstract-- i wondered who the exercise was for, but later in the description it was clear that it was for the patient. All these were about training the caregiver to care for the patient. There is no assessment of caregiver's needs.

Introduction-- leads to the need for the study.

Methods-- Qualitative and quantitative methods were described in enough detail that the study could be replicated. It would have been good to have The Intervention Acceptability Scale reference where it was introduced and for transparency that it was an author developed scale.

Results ---Well written. Desirability may have something to do with the high scores. The qualitative results were through and resonated with what we hear from rural caregivers.

Discussion-- I always look for something that engages the reader as if they are having a discussion with the authors. I also want to see the results discussed in light of the other research. Both were well done.

Limitations---Yes only being done is one province is a limitation, but in my view a minor one. There seem to be ceiling effects on the 4 point scale and likely a desirability effect.

Conclusions --- aligned with findings and relevant

References -- recent and well selected.

All in all, nicely done

Reviewer #2: This study examined perceived acceptability of four interventions for post-discharge care to rural caregivers in Canada. The authors found acceptability was high. However, family caregivers stated that the physical activity intervention would be challenging to implement. The study was well done, and findings were interesting.

1) The authors should expand the reflexivity section of their manuscript. They only mentioned the interviewer's gender. They should also talk about other socio-economic factors that could affect interactions between the interviewer and participants. People who wrote the manuscript and those who analyzed the data should also be included in the reflexivity section.

2) Was the Intervention Acceptability Scale validated for a rural population in Canada?

3) Caucasian is an outdated term [Flanagin A, Frey T, Christiansen SL; AMA Manual of Style Committee. Updated Guidance on the Reporting of Race and Ethnicity in Medical and Science Journals. JAMA. 2021;326(7):621-627]. The authors should use white.

4) The authors could do ANOVA tests for their descriptive table.

5) Instead of low levels of education, the authors could use limited formal education

6) In the discussion section, the authors should also acknowledge that their findings were not generalizable to racially/ethnically minoritized groups, and people who do not speak English.

7) The authors should have an appendix for their semi-structured interview questions.

6. PLOS authors have the option to publish the peer review history of their article (what does this mean?). If published, this will include your full peer review and any attached files.

Reviewer #1: No

Reviewer #2: No

---

## [Author Response · Author response to Decision Letter 0]

14 Oct 2022

Journal Requirements

Comment: Please ensure that your manuscript meets PLOS ONE's style requirements, including those for file naming.

Response: We have reviewed the requirements to ensure the manuscript meets PLOS ONE's style requirements.

Comment: We note that you have indicated that data from this study are available upon request. PLOS only allows data to be available upon request if there are legal or ethical restrictions on sharing data publicly. For more information on unacceptable data access restrictions, please see http://journals.plos.org/plosone/s/data-availability - loc-unacceptable-data-access-restrictions. 

Response: According to our participating hospital Research Ethics Board (REB), we are not permitted to post the study data for public consumption, and we do not have the authority to share our data because our sample was recruited from small towns and participants’ narratives may be recognizable to others even in de-identified data. However, researchers interested in using our data may submit a request to the hospital REB to request use of the data.

Contact info:

Research Ethics Board

Health Sciences North Research Institute

56 Walford Road

Sudbury, ON, P3E 2H2

Phone: 705-523-7300

Email: reb@hsnsudbury.ca

Comment: Please amend your list of authors on the manuscript to ensure that each author is linked to an affiliation. Authors’ affiliations should reflect the institution where the work was done (if authors moved subsequently, you can also list the new affiliation stating “current affiliation:….” as necessary).

Response: The authors’ affiliations have been amended so that each author is linked to an affiliation.

Comment: Your ethics statement should only appear in the Methods section of your manuscript.

Response: We have relocated out ethics statement to the methods section of the manuscript.

Comment: Please include captions for your Supporting Information files at the end of your manuscript, and update any in-text citations to match accordingly

Response: We have added Supporting Information Captions to the end of the manuscript per journal requirements.

Comment: Please review your reference list to ensure that it is complete and correct. If you have cited papers that have been retracted, please include the rationale for doing so in the manuscript text, or remove these references and replace them with relevant current references. Any changes to the reference list should be mentioned in the rebuttal letter that accompanies your revised manuscript. 

Response: We have reviewed the reference list to ensure it is complete and accurate. The following five new references have been added to the manuscript; they appear in red font so that they are easily recognizable:

1. O'Rourke HM, Sidani S, Jeffery N, Prestwich J, McLean H. Acceptability of personal contact interventions to address loneliness for people with dementia: An exploratory mixed methods study. IJNS Advances. 2020;2:100009.

2. Fox MT, Sidani S, Brooks D, McCague H. Perceived acceptability and preferences for low-intensity early activity interventions of older hospitalized medical patients exposed to bed rest: A cross sectional study. BMC Geriatr. 2018;18(53).

3. Sidani S, Epstein D, Fox MT, Miranda J. Psychometric Properties of the Treatment Perception and Preferences Measure. Clin Nurs Res. 2018;27:743-61.

4. Fox MT, Sidani S, Zaheer S, Butler JI. Healthcare consumers' and professionals' perceived acceptability of evidence-based interventions for rural transitional care. Worldviews Evid Based Nurs. 2022.

5. Sidani S, Epstein D, Miranda J. Eliciting patient treatment preferences: A strategy to integrate evidence-based and patient-centered care. Worldviews Evid Based Nurs. 2006;3(3):116-23.

Reviewer 1

Comment: Abstract-- i wondered who the exercise was for, but later in the description it was clear that it was for the patient. All these were about training the caregiver to care for the patient. There is no assessment of caregiver's needs.

Response: We did not formally assess family caregivers’ needs using a quantitative scale. However, in the qualitative interviews, we asked participants to identify their needs (please refer to the interview guide that is now appended to the paper) as a segway to indicate whether or not the interventions addressed their needs. In this paper, we focus on the latter point, that is, whether the interventions meet their needs.

The interview questions inquiring about needs include:

- "Did they (healthcare providers) ask about what you would need to manage your relative’s care and recovery at home after discharge from the hospital"

- "At any point since coming home, did you need more help in managing your relative’s care and recovery?"

- "Summary and other needs: Are there any other things you have found difficult and unprepared for now that your relative is back home that you think are important?" 

Comment: It would have been good to have The Intervention Acceptability Scale reference where it was introduced and for transparency that it was an author developed scale.

Response: In the design section, we now reference the Intervention Acceptability Scale. In the methods section, we have expanded our description of the Intervention Acceptability Scale to clarify that we contextualized the scale for each intervention, as recommended by the scale developer. 

Comment: Desirability may have something to do with the high scores. There seem to be ceiling effects on the 4-point scale and likely a desirability effect.

Response: We have considered this as a potential limitation. We indicate that although the Intervention Acceptability Scale did not manifest ceiling effects in prior research, it is possible that the scale demonstrated this effect in the current study.

Reviewer 2

Comment: The authors should expand the reflexivity section of their manuscript. They only mentioned the interviewer's gender. They should also talk about other socio-economic factors that could affect interactions between the interviewer and participants. People who wrote the manuscript and those who analyzed the data should also be included in the reflexivity section.

Response: We have expanded our discussion of reflexivity in both the data collection and analysis sections of the manuscript.

Comment: Was the Intervention Acceptability Scale validated for a rural population in Canada?

Response: We now explain that the scale has been used with different health interventions and populations and that the scale has demonstrated internal consistency reliability, as well as content, discriminant and factorial validity in these prior research studies. To the best of our knowledge, no other studies have validated the measure for rural populations in Canada, but we explain that in the larger study with rural participants, the scale demonstrated internal consistency reliability across the four interventions.

Comment: Caucasian is an outdated term. The authors should use white.

Response: We have changed Caucasian to white.

Comment: The authors could do ANOVA tests for their descriptive table.

Response: We now indicate in the table that there were no statistically significant differences between the average acceptability scale or subscale scores of the 4 interventions. 

Comment: Instead of low levels of education, the authors could use limited formal education

Response: We have changed this term to limited formal education.

Comment: In the discussion section, the authors should also acknowledge that their findings were not generalizable to racially/ethnically minoritized groups, and people who do not speak English.

Response: We now acknowledge this in the discussion section.

Comment: The authors should have an appendix for their semi-structured interview questions.

Response: We now include the semi-structured interview guide as Supporting Information File_S2.

Comment (suggestion contained in notes in the body of the manuscript): Family caregivers should be defined.

Response: We now defined family caregivers in the introduction section. We also explain that friends were eligible but that all the participants identified as family members.

---

## [Editor Report · Decision Letter 1]

1 Dec 2022

Family caregivers’ perspectives on the acceptability of four interventions proposed for rural transitional care: A multi-method study

PONE-D-22-12531R1

Dear Dr. Fox,

We’re pleased to inform you that your manuscript has been judged scientifically suitable for publication and will be formally accepted for publication once it meets all outstanding technical requirements.

Kind regards,

Dr Joseph Donlan

Senior Editor

PLOS ONE
---

## [Editor Report · Acceptance letter]

12 Dec 2022

PONE-D-22-12531R1 

Family caregivers’ perspectives on the acceptability of four interventions proposed for rural transitional care: A multi-method study 

Dear Dr. Fox:

I'm pleased to inform you that your manuscript has been deemed suitable for publication in PLOS ONE. Congratulations! Your manuscript is now with our production department. 

Kind regards, 

on behalf of

Dr Joseph Donlan 

Staff Editor

PLOS ONE